# Satellite Navigation Spoofing Interference Detection and Direction Finding Based on Array Antenna

**DOI:** 10.3390/s23031604

**Published:** 2023-02-01

**Authors:** Huiyun Yang, Ruimin Jin, Wenpu Xu, Lei Che, Weimin Zhen

**Affiliations:** 1China Research Institute of Radiowave Propagation, Qingdao 266107, China; 2National Key Laboratory of Electromagnetic Environment, Qingdao 266107, China; 3School of Electronic Engineering, Xidian University, Xi’an 710071, China

**Keywords:** GNSS, array antenna, spoofing interference, detection, direction finding

## Abstract

Satellite navigation signals are feeble when they reach the ground, so they are vulnerable to attacks from outside interference signals. By emitting spoofing interference signals similar to real satellite signals, spoofing interference can make receivers give wrong navigation, position, and time information, and it is challenging to detect. This seriously affects the safe use of GNSS; therefore, it is essential to identify spoofing interference signals quickly and accurately. In our study, we proposed a novel six-array spoofing-interference-monitoring array antenna, which achieved the detection and identification of spoofing interference sources by monitoring the relevant peaks and combining an airspace-trapping algorithm. Moreover, we quickly accomplished our search for the whole circumferential ambiguity by using long- and short-baseline algorithms, which can realize the high-precision detection of spoofing interference sources. To verify this method’s accuracy, we conducted outdoor real experiments using a special spoofing interference source, and our experimental results show that our proposed array antenna’s directional accuracy for spoofing interference signals is kept within 2°, showing high spoofing interference direction-finding capability.

## 1. Introduction

Global navigation satellite systems (GNSS) provide users with all-weather, high-precision position, velocity, and time (PVT) services, which are widely used in national economic and military fields, i.e., transportation, electricity, finance, telecommunications, military aviation, precision-guided weapons, etc. Because of the importance of satellite navigation systems to the security and economic development of each country, all major powers in the world are actively striving to develop their satellite navigation systems, which mainly include the GPS, Galileo, GLONASS, and Beidou systems of the United States, Europe, Russia, and China, respectively [1,2,3,4].

As GNSS is widely used in various fields, its radio magnetic environment is becoming increasingly complex, making the satellite signal in the transmission process extremely vulnerable to intentional or unintentional interference, which can seriously affect the satellite navigation system’s ground monitoring, as well as the normal use of GNSS by users, posing a severe threat to the accuracy, availability, and integrity of satellite navigation. Unintentional interference is usually the result of electronic equipment signal leakage or natural phenomena, including radio frequency interference, ionospheric scintillation, the multipath effect, and non-line of sight (NLOS) propagation [5], etc. Intentional interference, including suppression and spoofing interference, usually disrupts normal satellite communications, forges spoofing interference signals, and generates suppression signals to block and deceive ground receivers so that the receiver obtains incorrect information [6]. Suppression jamming is performed by high-power-emitting signals so that the real satellite signal is drowned in the interference signal in terms of power or frequency band, meaning the navigation receiver cannot work normally. Spoofing interference techniques are relatively complex and are divided into transponder and generative spoofing interference categories. Among them, transponder spoofing interference through the spoofing interference source to receive the real signal modifies the real signal and then sends it out, while generative spoofing interference requires understanding the satellite structure to forge the satellite signal. Both ways deceive the navigation user so that they cannot be correctly positioned. Due to its particularity, the military signals in satellite navigation usually have confidentiality in the signal generation method and signal format, meaning the outside world cannot know the satellite signal parameters in advance; therefore, there are certain technical difficulties when attempting satellite military signal interference [7]. However, considering the extensiveness and convenience of civil satellite signals, the modulation mode and signal format of satellite signals are usually in an open state, making civil satellite signals extremely susceptible to various forms of interference [8,9].

Although the technology involved in spoofing interference is relatively complex, its harm is apparent. Currently, GNSS civil navigation signals are more susceptible to interference by GNSS navigation spoofing signal generators because they are not encrypted like military signals. This causes the navigation receiver to acquire misplaced position information under normal working conditions; therefore, spoofing interference must be detected, weakened, or eliminated.

In 2016, Lee et al. used the signal quality monitor (SQM) algorithm for spoofing interference detection based on the characteristics of spoofing signals that distort the correlation peaks [10]; in 2017, Margarita et al. studied the characteristics related to spoofing interference and improved the receiver of the Galileo system to achieve the function of anti-spoofing interference [11], but the method is only theoretically applicable to some civil systems, and there is no special equipment; in 2018, Geng proposed that the receiver could estimate the signal power before dispreading its received signal [12] to realize the detection of the spoofing signal, which can significantly reduce the amount of target-receiver computation—its implementation is simple but the applicability of the spoofing scenario is not high. In 2019, Zhang et al. exploited the fact that transposed spoofing interference is a delay to the real satellite signal by combining the observation of the number of correlation peaks with measuring the half-height width size to achieve the detection of spoofing interference with large and small time delays [13]. In 2020, Li et al. proposed a convolutional neural network-based method to effectively detect the presence of small-delay spoofing interference signals, with the method achieving high detection accuracy for small-delay spoofing interference signals [14]. In 2022, Wang et al. used the power combined with the SQM (PCS) method to detect spoofing interference signals and improve detection timeliness [15].

Aiming at the problem that the traditional spoofing interference detection method cannot locate the direction of the spoofing interference source, many scholars proposed the use of an array antenna to locate the direction of the spoofing interference source, drawing on the relevant interferometer direction measurement principle; using the same signal received by different array elements of the carrier phase difference to build the observation, it can realize the direction of the signal captured by the receiver. Spoof interference detection and identification can be achieved by comparing the satellite direction with the ephemeris solution. However, their methods remain in the experimental simulation stage and are not verified in the real outdoor environment. In the real environment, there will be interference from the external environment such as multipath, which has some influence on the detection of spoofing interference signals.

In our study, an L-shaped six-array antenna was specially designed for spoofing interference. In the real environment, the method we designed can realize the simultaneous capture and tracking of the normal and interference signals of the same satellite through multimodal monitoring and effectively separate the real satellite and spoofing interference signals by using the airspace identification method, followed by the long- and short-baseline interferometric-direction-finding method of the spoofing interference source to locate the direction of the interference source through the designed antenna array.

## 2. Materials and Methods

### 2.1. Spoofing Interference Sources

Spoofing interference, including transponder and generation interference, uses signal acquisition equipment to capture and amplify spoofing interference signals for directly broadcasting. Generative interference, on the other hand, simulates the generation of pseudosatellite signals and broadcasts them according to the institutional requirements of the navigation signal, which is more difficult to achieve compared with transponder interference.

Transponder spoofing interference is the direct transponder of the real navigation signals by the spoofing interference sources. Figure 1 shows that after delay and power amplification, S1, S2, S3, and S4 denote satellites, J denotes the position of the spoofing interference sources, R denotes the position of the normal receiver, and F denotes the position to which the receiver is spoofed to locate. The spoofing interference source generates a spoofing signal by forwarding the real signal without generating a high-fidelity interference signal; the spoofing interference signal is captured by receivers within a certain range, which is relatively easy to implement technically. Furthermore, its power is stronger than the real signal, meaning there is a high probability of being captured. Theoretically, successful spoofing is a probabilistic event.

A spoofing interference source is placed at *J*. In the interference effective region, the spoofed receiver is assumed to be located at *R*. If the spoofing destination is *F*, then:(1)$$\begin{array}{c} \begin{array}{c} \rho_{1}=S_{1}F+ct_{R}\\ \rho_{2}=S_{2}F+ct_{R}\\ \rho_{3}=S_{3}F+ct_{R}\\ \rho_{4}=S_{4}F+ct_{R}\;\; \end{array} \end{array}$$ If $\rho_{i}$ (i=1,2,3,4) is the observation pseudorange, $S_{i}f$ (i=1,2,3,4) is the distance from the satellite to the spoofing point, and *c* is the speed of light, $t_{R}$ is the receiver clock difference (atmospheric delay is neglected in the equation), and the equation when the receiver at the *R* is subjected to the transponder spoofing interference signal is:(2)$$\begin{array}{c} \begin{array}{c} \begin{array}{c} \rho_{1}^{\prime }=S_{1}J+JR+ct_{1}+ct_{R}\\ \rho_{2}^{\prime }=S_{2}J+JR+ct_{2}+ct_{R}\\ \rho_{3}^{\prime }=S_{3}J+JR+ct_{3}+ct_{R}\\ \rho_{4}^{\prime }=S_{4}J+JR+ct_{4}+ct_{R} \end{array} \end{array} \end{array}$$

If $\rho_{i}^{\prime }$ (i=1,2,3,4) is the observation pseudorange of the transponder, $S_{i}J$ (i=1,2,3,4) is the distance from the satellite to the spoofing interference source, JR is the distance from the spoofing interference source to receiver, *c* is the speed of light, $t_{i}$ is the delay time to the signal of the satellite, $t_{R}$ is the receiver clock difference (atmospheric delay is neglected in the equation), let $\rho_{i}=\rho_{i}^{\prime }$, according to the pseudorange positioning principle of the GNSS, Equations (1) and (2) will solve the same position, and $S_{i}F=S_{i}J+JR+ct_{i}$. Because the physical delay needs to be positive, it is required that $JR<S_{i}F-S_{i}J$, and when the timing is accurate and does not affect the receiver clock difference, assuming that the condition is not satisfied, then
(3)$$\tau =\frac{\min(S_{i}F-S_{i}J-JR)}{c}$$

The delay correction is $t_{i}^{\prime }=t_{i}+\left|\tau \right|$, and the receiver clock difference resulting from the substitution in Equation (3) is $t_{R}^{\prime }=t_{R}+\left|\tau \right|$. The receiver clock difference has a jump, which is not what the spoofers would like to see. This is the disadvantage of transponder spoofing interference, which puts a high demand on the real-time performance of the spoofing interference sources.

Generative spoofing interference is the premise that the interferer does not rely on the real signal and independently emits a spoofing signal similar to the real signal. Through the relative motion of the peaks associated with the two, the spoofing peak gradually strips the real peak from the tracking loop by virtue of its power advantage, and then controls the tracking loop [16,17]. Figure 2 shows the generative spoofing interference flow.

GNSS spoofing interference signals typically include the following characteristics:
(1)The frequency/pseudocode sequence of the interference signal is the same as the normal GNSS signal;(2)There is a difference between the phase of the interference signal pseudocode and the normal GNSS signal, usually greater than 1.5 code chips;(3)A single spoofing interference source usually broadcasts more than 4 GNSS satellite spoofing interference signals simultaneously.

When there is a GNSS spoofing interference source around the GNSS receiver, the GNSS receiver will directly capture the peak position related to the spoofing interference signal in the satellite capture stage under the condition that the relevant peak’s maximum determination criterion is adopted, and the spoofing interference signal will be stably tracked through the tracking loop. When each channel of the GNSS receiver is stably tracked to the spoofing interference signal, the receiver positioning solution result is completely controlled by the GNSS spoofing interferer. To realize the monitoring and effective identification of GNSS interference signals, the GNSS receiver can monitor according to the relevant peak morphology of each satellite and deceive the interference and identification of real satellite signals through the airspace characteristics of real satellite and interference signals [18,19].

To facilitate the experiment, this paper uses a self-researched spoofing interference source. The spoofing interference source can simultaneously send the spoofing navigation signal of the three GPS L1/BDS B1I/BDS B1C frequency points, while the navigation signal power and pseudocode can be set as needed, and the power of the spoofing interference signal is continuously adjustable within −135~−60 dBm. Furthermore, our methods are verified according to the test requirements.

Figure 3 shows the flow of spoofing interference detection.

### 2.2. Spoofing Interference Detection

In the absence of interference, taking the GPS L1 navigation signal as an example, the satellite signal received by the receiver is
(4)$$s\left(t\right)=c_{i}\left(t\right)\times d_{i}\left(t\right)\cos(2\pi \left(f_{i}+\Delta f_{i}\right)t+\theta )$$
where $c_{i}\left(t\right)$ is the spread spectrum code sequence of the *i*-th satellite, $d_{i}\left(t\right)$ is the navigation message information of the *i*-th satellite, $f_{i}$ is the center frequency of the *i*-th satellite, $\Delta f_{i}$ is the Doppler shift of the *i*-th satellite, and θ is the initial phase of the *i*-th satellite. After the receiver is synchronized through the captured trace, the receiver can ideally obtain a navigation signal correlation function, which is:(5)$$R\left(\tau \right)=1-\left|\tau \right|/T_{c}$$
where $R\left(\tau \right)$ is the baseband correlation function after the accurate stripping of carrier Doppler, τ is the delay of the initial phase of the local spread spectrum code and the phase difference between the received navigation signals, and $T_{c}$ is the spread spectrum code period of the navigation signal.

In the presence of spoofing interference, the satellite signal received by the receiver is:(6)$$s_{m}\left(t\right)=A_{i}\left(t\right)c_{i}\left(t\right)\times d_{i}\left(t\right)\cos(2\pi \left(f_{i}+\Delta f\right)t+\theta )+\overset{n}{\underset{k=1}{\sum }}B_{k}\left(t\right)c_{i}\left(t+t_{k}\right)\times d_{i}\left(t+t_{k}\right)\cos(2\pi \left(f_{i}+\Delta f_{k}\right)t+\theta_{k})$$
where $s_{m}\left(t\right)$ is the navigation signal received by the receiver containing spoofing interference, $c_{i}\left(t\right)$ represents the spread spectrum code of the ith satellite, $d_{i}\left(t\right)$ represents the navigation message data of the ith satellite, $t_{k}$ represents the time difference between the initial pseudocode phase between the spoofing interference signal and the real signal, $\Delta f_{k}$ represents the Doppler frequency offset of the spoofing interference signal, $A_{i}\left(t\right)$ represents the amplitude of the real navigation signal, $B_{k}\left(t\right)$ represents the amplitude of the spoofing interference signal, and k represents the existence of k spoofing interference source signals for the i-th navigation satellite. The receiver navigation signal correlation peak function can be expressed as:(7)$$R_{m}\left(\tau \right)=A_{i}^{2}\left(1-\left|\tau \right|/T_{c}\right)+\overset{n}{\underset{k=1}{\sum }}B_{k}^{2}\left(1-\left|\tau -t_{k}\right|/T_{c}\right)\cos(\theta_{k}-\theta )$$

The detection of multi-correlated peaks can be expressed as the following ternary hypothesis problem: satellite signal correlated peaks and spoofing signal correlation peaks and no navigation signal.
(8)$$\left\{\begin{array}{c} H_{0}:x_{n}=W_{n}\\ H_{1}:x_{n}=S_{n}+W_{n}\\ H_{2}:x_{n}=S_{n}^{\left(sp\right)}+W_{n} \end{array}\right.$$
where $W_{n}$ is additive white noise, $S_{n}$ is the real satellite signal, $S_{n}^{\left(sp\right)}$ is the spoofing interference signal, and $x_{n}$ is the detection amount. $H_{0}$, $H_{1}$, and $H_{2}$ indicate that the navigation signal does not exist, the navigation signal exists, and the spoofing signal exists, respectively. Based on the real satellite signal detection threshold $V_{t}$, a threshold $V_{s}$ is introduced when the number of relevant peaks exceeding the detection threshold $V_{t}$ is more than two and the relevant peak exceeds the threshold $V_{s}$, while the signal corresponding to the relevant peak is considered to be a spoofing signal. The relationship between the detection statistic V and thresholds $V_{t}$ and $V_{s}$ can be expressed as:(9)$$\left\{\begin{array}{c} H_{0}:V<V_{t}\\ H_{1}:V_{t}\leq V<V_{s}\\ H_{2}:V\geq V_{s} \end{array}\right.$$

Assuming $H_{1}$ and $H_{2}$, the signal strength is $\lambda \gg \sigma_{n}^{2}$, in which case the mean of the detection statistic V is approximately expressed as:(10)$$E\left(V\right)=\sqrt{\frac{\pi }{2}}\sigma_{n}L_{\frac{1}{2}}\left(-\frac{\lambda^{2}}{2\sigma_{n}^{2}}\right)\approx \lambda$$
where $\sigma_{n}$ is the noise intensity, and the Lagrangian polynomial $L_{\frac{1}{2}}$ is:(11)$$L_{\frac{1}{2}}\left(x\right)=e^{\frac{x}{2}}\left[\left(1-x\right)I_{0}\left(-\frac{x}{2}\right)-xI_{1}\left(-\frac{x}{2}\right)\right]$$
where $I_{0}$ and $I_{1}$ are zero- and first-order modified Bezier functions, respectively.

Because navigation and spoofing signals are independently distributed, their probability density functions are:(12)$$f\left(v,\lambda_{1}\right)=\frac{v}{\sigma_{n}^{2}}e^{-\frac{v^{2}+\lambda_{1}^{2}}{2\sigma_{n}^{2}}}I_{0}\left(\frac{\lambda_{1}v}{\sigma_{n}}\right)$$
(13)$$f\left(v,\lambda_{2}\right)=\frac{v}{\sigma_{n}^{2}}e^{-\frac{v^{2}+\lambda_{2}^{2}}{2\sigma_{n}^{2}}}I_{0}\left(\frac{\lambda_{2}v}{\sigma_{n}}\right)$$
where v is the detection statistic, $\lambda_{1}$ is the mean of the incoherent integration of the navigation signal, $\lambda_{2}$ is the mean of the incoherent integration of the spoofing interference signal, and $\lambda_{2}>\lambda_{1}$.

The spoofing signal detection false-alarm probability can be expressed as:(14)$$P_{fa}=\int_{V_{s}}^{\infty }f\left(v,\lambda_{1}\right)dv=\int_{V_{s}}^{\infty }\frac{v}{\sigma_{n}^{2}}e^{-\frac{v^{2}+\lambda_{1}^{2}}{2\sigma_{n}^{2}}}I_{0}\left(\frac{\lambda_{1}v}{\sigma_{n}}\right)dv$$
where v is the detection statistic, and $V_{s}$ is the set spoofing interference detection threshold.

Therefore, given the noise power $\sigma_{n}^{2}$, false-alarm probability $P_{fa}$, and mean $\lambda_{1}$ of the non-coherent integral of the navigation signal, the spoofing signal power detection threshold $V_{s}$ can be calculated, and the spoofing interference signal can be detected by the multimodal detection of the capture module.

### 2.3. Spoofing Interference Identification

Considering that the multipath of a single satellite of the receiver will also introduce a multimodal environment, which will lead the receiver to recognize false alarms for fraudulent interference [20], based on multimodal detection, we used the multi-antenna array to zero the notch to identify the wave arrival angle of all satellites [21,22,23,24,25].

The navigation signal received by the GNSS array antenna enters the system through the antenna array composed of M antennas, as shown in Figure 4, and the data are converted into an MN channel after delay processing, which is represented by the symbol X. W represents the processor weight vector (MN × 1 dimension), and R = E[XX^H^] represents the covariance matrix (MN× MN dimension) of the received data.

Zeroing weights for a certain direction is equivalent to a constrained optimization problem:(15)$$w{}_{s}^{opt}=\arg\;\min\;w{}^{H}Rw,\;w^{H}S=1$$
where *S* is the space-time two-dimensional guide vector, *S* = *S*_S_⊗*S*_T_, where ⊗ is the Kroeker product, $S_{S}={\left[1,e^{jw_{s}},\cdots e^{j\left(M-1\right)w_{s}}\right]}^{T},\;$ and $S_{T}={\left[1,e^{jw_{t}},\cdots e^{j\left(N-1\right)w_{t}}\right]}^{T}$. $w_{s}=\frac{2\pi d\sin\theta }{\lambda }$ and $w_{t}=2\pi fT_{s}$, where d is the distance between two adjacent antennas in the antenna array, θ is the angle of incidence, that is, the angle between the direction of incidence and the normal direction of the array, and λ is the wavelength. The solution of the optimal space-time processor is:(16)$$\begin{array}{c} w_{s}^{\mathrm{opt}\;}={\left[S^{H}R^{-1}S\right]}^{-1}R^{-1}S\left(\mathrm{MN}\times 1\right) \end{array}$$

Figure 5 shows the schematic of airspace trap. If the receiver finds that some satellites have multimodal characteristics, it can complete the zero-trapping direction of the received signal by zero-trap scanning in the whole airspace, and when more than four satellites lose signals or the carrier-to-noise ratio is greatly reduced in a certain direction, the existence of a spoofing interference signal is determined and the direction of the incoming wave of the spoofing interference signal is obtained.

### 2.4. Spoofing Interference Source Direction Finding

In this study, we adopted the long- and short-baseline interferometric-direction-finding method for the direction finding of the deceptive interference source, and the direction-finding accuracy is related to the array antenna array [26,27], while the designed array antenna configuration is as follows in Figure 6:

The direction-finding accuracy of the spoofing interferer can be obtained by referring to the interferometric goniometric formula, and the specific calculation steps are as follows:

(1) The baseline lengths of antennas 1 and 2 are the half wavelength, and we used the long- and short-baseline algorithm to remove the integer ambiguity of the carrier phase spoofing interference source.

(2) The angular accuracy is directly related to the incidence angle and carrier phase tracking accuracy, which can be calculated by index decomposition through the engineering experience formula.
(17)$$\Delta \theta =\frac{\lambda \Delta \varphi }{2\pi d\cos\theta }+\frac{\Delta \lambda }{\lambda }\tan\theta$$
where λ is the GNSS signal wavelength, d is the antenna baseline length, Δφ is the carrier phase tracking error, θ is the incidence angle of the spoofing interference source, and Δλ is the wavelength measurement error introduced by Doppler.

In the above equation, interferometric goniometry adopts a third-order PLL carrier tracking loop, and the wavelength measurement error introduced by the carrier Doppler is negligible when the receiver is working stably; therefore, the angle measurement error formula can be simplified to
(18)$$\Delta \theta =\frac{\lambda \Delta \varphi }{2\pi d\cos\theta }$$
where Δφ is the amount of PLL carrier phase measurement error after the stable tracking of the receiver carrier tracking loop, which is directly related to the signal-to-noise ratio of the spoofing interferer signal and the receiver phase noise; and after receiving the stable tracking spoofing interferer signal, the carrier phase measurement accuracy can be expressed as:(19)$$\sigma_{PLL}=\frac{180^{{}^{\circ}}}{\pi }\sqrt{\frac{B_{L}}{C/N_{0}}\left(1+\frac{1}{2T_{coh}C/N_{0}}\right)}$$
where $B_{L}$ is the loop noise bandwidth, $C/N_{0}$ is the carrier-to-noise ratio, and $T_{coh}$ is the pre-detection integral time.

(3) Using a right-angle interference array, after simulation analysis, under the condition that the carrier phase measurement error is fixed, the single-baseline angle measurement accuracy is approximately 11.5° when the incidence angle is 50°, and the angular measurement accuracy is 8° at the incidence angle of 22°. Considering that the system adopts a vertical double baseline for direction finding, the baseline angle is 90°, and when the incidence angle is >45°, the vertical baseline can be used to complete the direction finding of the interference source, while the short-baseline direction-finding accuracy is 13° (RF-channel zero value and antenna installation error factors).

(4) To ensure the accuracy of direction finding of deceptive interference sources, our use of antennas 1, 2, 3, 5, and 6 involved vertical double baselines to achieve the accurate direction finding of deceptive interference sources, while the lengths of the long and short baselines were λ/2 and 2λ, respectively. The long baseline in this mode can quickly complete the whole week fuzzy search through short-baseline assistance, realizing the high-precision direction-finding function.

After simulation analysis, under the condition that the carrier phase measurement error is fixed, the single-baseline angle measurement accuracy is about 6° at 75°, and 2.2° at 45°. Considering that the system adopts vertical double-baseline direction finding, the baseline angle is 90°, and when the incidence angle is 45°, the vertical baseline can be used to complete the direction finding of the interference source, and the accuracy of the long baseline angle measurement is 1.3°.

## 3. Results and Discussion

For a single spoofing interference source, we used our direction-finding method to establish an outfield test environment. The test equipment included a spoofing interference source, transmitting antenna, set of high-precision directional receiver computers, power supply, set of RF cables, and spoofing interference detection and direction-finding device.

On a circumference with a deceptive interference detection and direction-finding antenna as the center and a radius of 30 m, a high-precision GNSS receiver was used to calibrate a point every 60 degrees, and 0 degrees represented the true north direction. The transmitting antenna of the interference source was placed on each calibration point in turn, the direction of the deceptive interference source was determined, and when the deceptive interference source transmitting antenna is erected, the incidence angle should be greater than 15 degrees by as much as possible.

Table 1 shows that, when the power of the spoofing signal is −124 dBm, the spoofing interference source simultaneously broadcasts the synchronous spoofing signal of satellites 6/8/13/9/16 and regenerates the spoofing signal of satellite 19. The two types of interference signals are simultaneously transmitted in one channel to realize spoofing interference to the regional receiver.

Figure 7 shows the correlation peaks in spoofing interference environment. This method can simultaneously capture and track the normal and interference signals of the same satellite through multi-peak monitoring and realize the multi-channel concurrent tracking of satellites 6/8/13/9/16 to complete the spoof interference monitoring. Therefore, the spatial identification method can be used to effectively separate real satellite signals from spoofing interference signals. Our test results show that channels 1–7, 9–13, 16, and 20 are tracking normal GPS signals, and channels 6, 14, 15, 17, 18, and 19 are spoofing interference signals, which is consistent with the signals sent by spoofing interference sources.

At the same time, we calculated the detection time of the spoofing interference. The average time from the signal source to the detection of spoofing interference is about 7.7 s, showing good real-time detection. Table 2 shows the detection-time statistics:

We used our novel vertical dual-baseline antenna array to achieve accurate direction findings of spoofing interference sources. In this mode, the long baseline can be assisted by the short baseline to quickly complete the whole-circle fuzzy search and realize the high-precision direction-finding function. The mean error accuracy is kept within 2° at approximately 1.4°. Table 3 shows the test’s direction-finding results.

Due to the existence of the antenna installation and RF-channel zero-calibration errors in the actual tests, there will also be complex multipath interference and other problems; therefore, the calibration of the electronic compass and cable will also have some impact on accuracy. The angle measurement accuracy error of this method is slightly lower than the simulation results. However, it can still achieve high direction-finding accuracy.

## 4. Conclusions

In our study, we detected the spoofing interference source by monitoring the correlation peaks and zeroing the spatial notch, which effectively reduces the false alarm caused by the multipath environment. At the same time, we designed an antenna array with vertical double baselines, which uses its long and short baselines to quickly complete the whole-circle ambiguity search and realize the high-precision direction finding of spoofing interference sources, with a directional accuracy within 2° and detection time of about 5 s. The designed method has high precision and strong feasibility, can be used as a reference for other navigation-frequency-band spoof detection and direction-finding methods, and has high theoretical and practical value. At present, this method has been applied to the equipment developed and verified in the test.

At present, there are still some problems to be solved, and future research can be carried out in the following aspects:(1)Although we verified the reliance on array antennas for spoofing interference detection and direction finding using a relatively simple electromagnetic environment through experiments, in a complex electromagnetic environment, the receiver will be affected by a variety of interference types, such as ionospheric scintillation, solar radio burst, suppression, and spoofing interference. At present, research on interference detection mostly focuses on one of the interferences alone, meaning that methods of unified identification and the classification of different interferences need to be urgently studied;(2)The traditional spoofing interference detection methods are limited by the model or need additional equipment, and the detection parameters are single, making it difficult to reflect the characteristics of spoofing interference signals comprehensively. In the future, deep-learning-based methods could be considered for studies of spoofing interference detection methods based on the existing receivers.

## Figures and Tables

**Figure 1 sensors-23-01604-f001:**
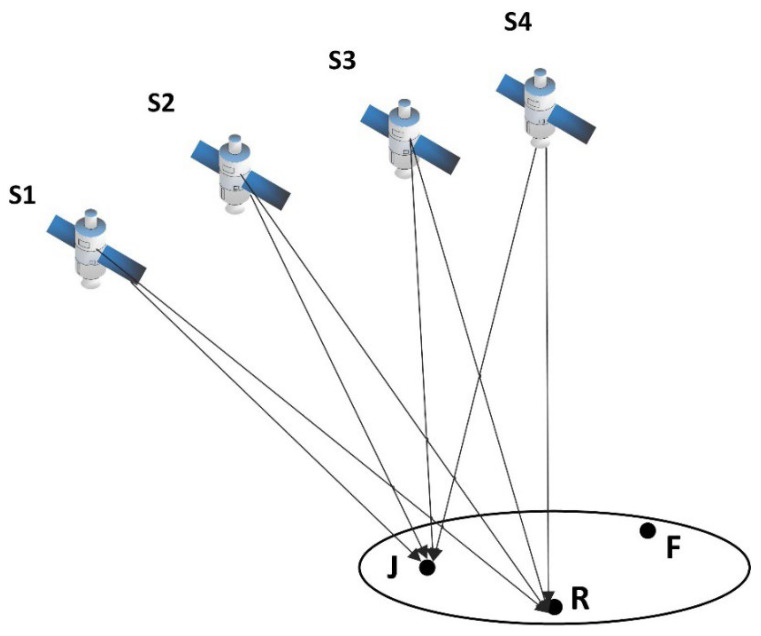
Transponder spoofing interference schematic.

**Figure 2 sensors-23-01604-f002:**
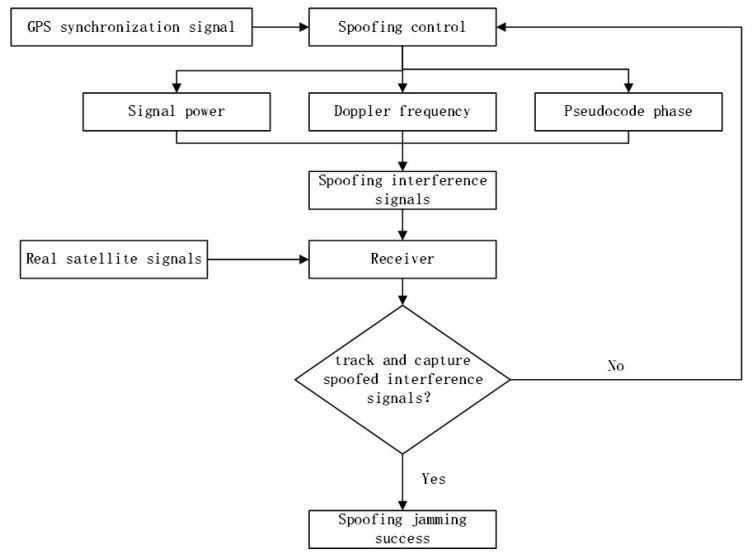
Generative spoofing interference process.

**Figure 3 sensors-23-01604-f003:**
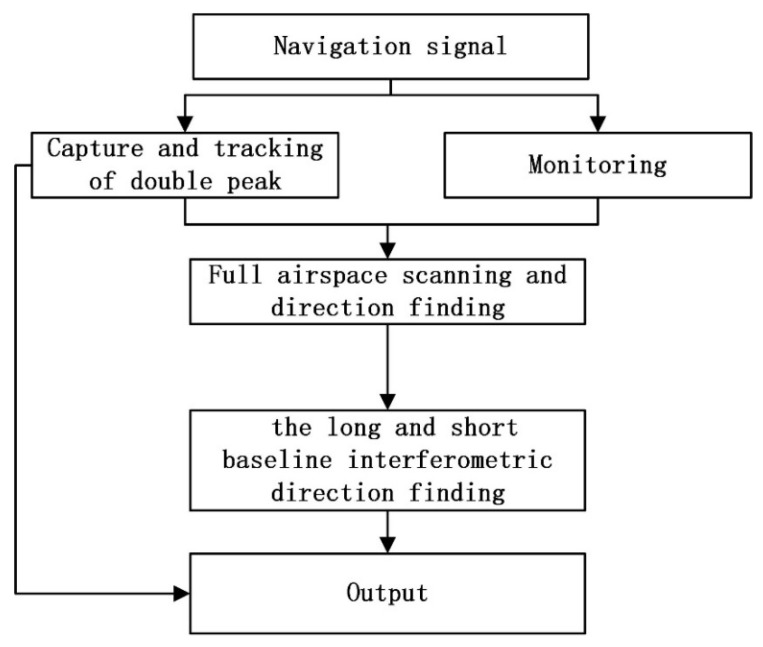
Spoofing interference detection process.

**Figure 4 sensors-23-01604-f004:**
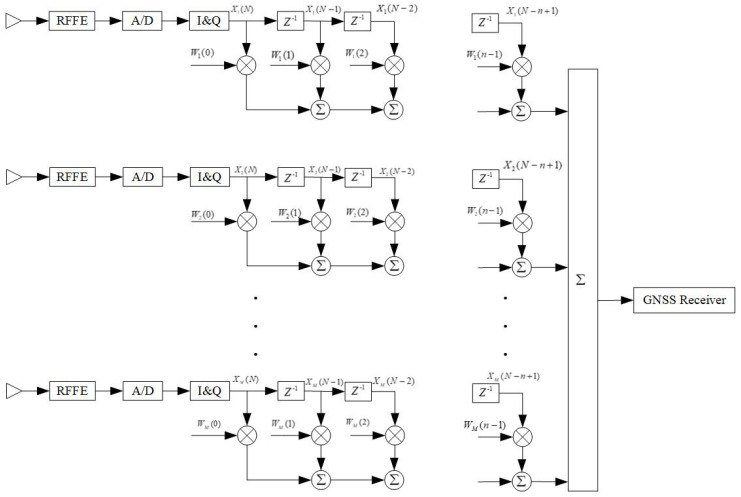
Space-time two-dimensional filtering algorithm.

**Figure 5 sensors-23-01604-f005:**
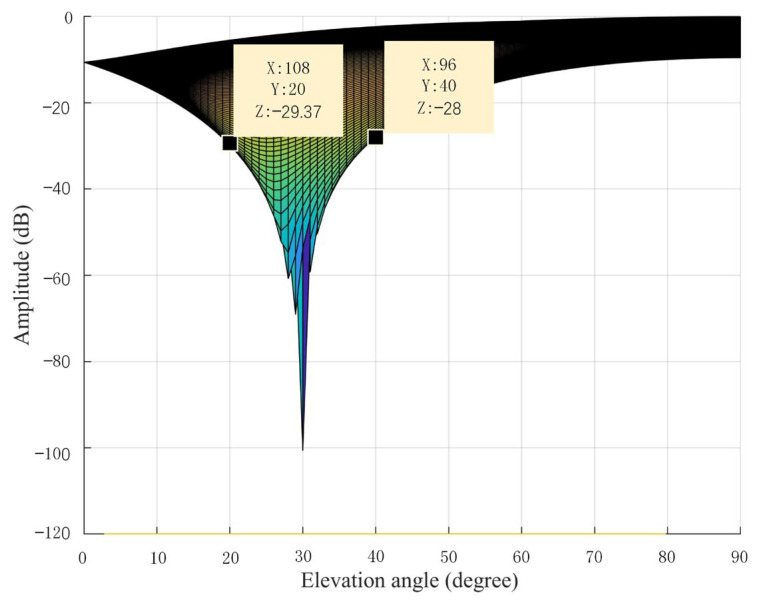
Zero airspace scanning beamwidth.

**Figure 6 sensors-23-01604-f006:**
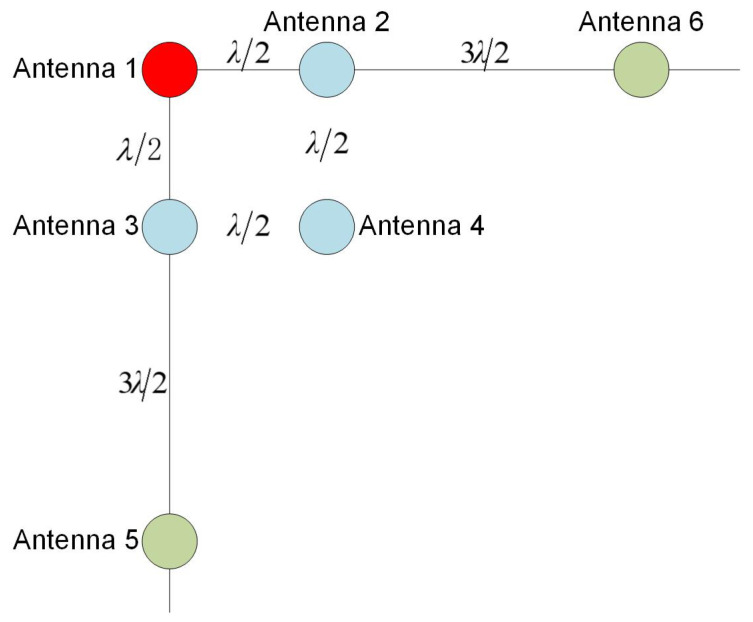
Array antenna configuration.

**Figure 7 sensors-23-01604-f007:**
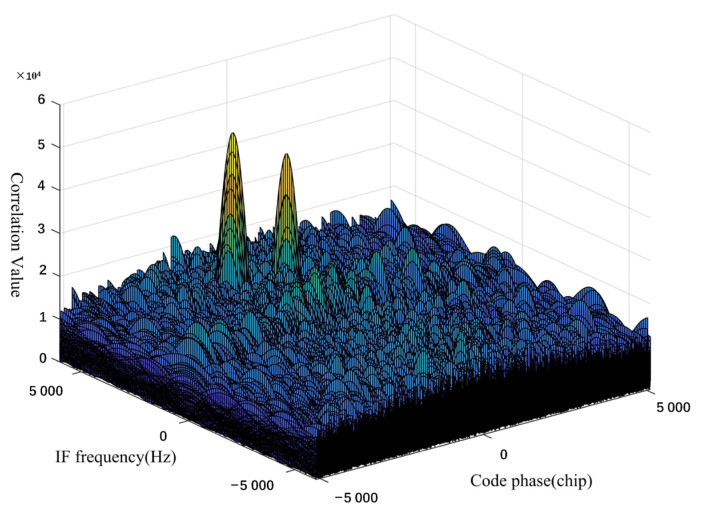
Schematic diagram of frequency-code two-dimensional correlation peaks in spoofing interference environment.

**Table 1 sensors-23-01604-t001:** Result of spoofing interference detection.

Channel	Satellite Number	Pseudorange (m)	C/N0	Track Status	Detection Result
1	6	36,273,924.8	40.9	Tracking	True
2	36	22,249,771.6	46.6	Tracking	True
3	13	36,668,067.6	42.3	Tracking	True
4	16	36,508,379.1	44.9	Tracking	True
5	2	37,236,339.6	41.2	Tracking	True
6	3	36,890,554.3	38.5	Tracking	True
7	26	26,477,344.4	44.2	Tracking	True
8	6	36,159,106.3	41.9	Tracking	Spoofing
9	9	36,766,094.4	39.7	Tracking	True
10	21	24,020,616.0	41.9	Tracking	True
11	22	21,789,099.7	45.5	Tracking	True
12	8	37,401,729.9	38.9	Tracking	True
13	8	36,992,831.9	41.3	Tracking	Spoofing
14	13	36,637,555.0	40.6	Tracking	Spoofing
15	19	23,981,686.4	35.9	Tracking	True
16	9	37,031,246.4	41.4	Tracking	Spoofing
17	19	24,176,708.4	40.6	Tracking	Spoofing
18	16	36,473,256.8	41.8	Tracking	Spoofing
19	5	38,484,892.8	36.4	Tracking	True

**Table 2 sensors-23-01604-t002:** Time of spoofing interference detection.

Number	1st	2nd	3rd	4th	5th	6th	7th	8th	9th	10th	Average
Time(s)	5.59	6.72	4.60	5.19	3.53	6.53	4.62	7.43	3.09	4.94	5.22

**Table 3 sensors-23-01604-t003:** Angle of direction finding.

Number	Ideal Angle	Actual Measurement Angle	Direction Angle	Relevance
1	0°	0.07°	0.60°	0.53°
2	60°	59.96°	58.10°	1.86°
3	120°	119.97°	121.45°	1.48°
4	180°	181.01°	183.44°	2.43°
5	240°	240.31°	241.10°	0.79°
6	300°	300.14°	301.53°	1.39°

## Data Availability

Not applicable.

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
