# Peer review of "Satellite Navigation Spoofing Interference Detection and Direction Finding Based on Array Antenna"

_sensors, 2023, doi:10.3390/s23031604_

Round 1

Reviewer 1 Report

The paper presents a GNSS spoofing detector based on statistical analysis of the received GNSS signals. The core of the paper is of interest, but unfortunately in my opinion it has to be deeply improved in order to be published. 

Some remarks:

L67. The comments of the Iranian engineer are not correctly quoted. Anyway, I suggest explaining the process based on Iranian sources, in place of reproducing the comments.

L108. The whole paragraph is hard to understand, even if it is a simple concept.
Fig1. Please identify the elements involved and use them to improve the explanation starting in L108.
L114: Repeated sentence.

English has to be reviewed, mainly in its sentence construction and length.

Eq1-2. What is ct_R? ... Is t_R the receiver clock difference?. I suppose that "c" is the speed of light in vacuum.
Eq1-2. \rho is the same in both equations; can you explain what is \rho_i' in L125?

L133. I imagine that "the device" is the "spoofing device" ... Please clarify.
In general, the explanation in section 2.1 about "spoofing interference sources" is confusing, but the concept is quite simple. Please improve the explanation and correct several grammar and variable name mistakes.

Fig2. The figure is bad, because the "No" derivation should be in the "track and capture ..." box. Anyway, in my opinion, the "No" derivation has to end with a "Spoofing jamming failure" box, in place to return to the "Spoofing control", because there are no return signals from the target to the "Jammer".

L194. The "following figure:" does not exist.

Equations 10-14 are not well explained and several terms are not described.

Fig4. It has a bad dimension and is not easy to distinguish.

L307. Please finish the paragraph about "what the test consists of" correctly.

In general, are the L2 or equivalent GNSS bands taken into account? and other question, are the systems using RTK affected by spoofing and jamming?. And in case of yes, is the solution proposed effective? In my opinion, several aspects of those questions have to be taken into account.

Reviewer 2 Report

Add a  description of the research background 

Author Response

Point 1:

Add a description of the research background.

Response 1:

Thank you for your considerate comments and suggestions. We have modified some descriptions of this article. Then, We added the following descriptions " Aiming at the problem of the traditional spoofing interference detection method cannot locate the direction of the spoofing interference source, many scholars proposed the use of array antenna to locate the direction of the spoofing interference source, drawing on the relevant interferometer direction measurement principle, using the same signal received by different array elements of the carrier phase difference to build the observation, it can realize the direction of signal captured by the receiver. Spoof interference detection and identification can be achieved by comparing the satellite direction with the ephemeris solution. However, their methods stay in the experimental simulation stage and are not verified in the real outdoor environment. In the real environment, there will be interference from the external environment such as multipath, which has some influence on the detection of spoofing interference signals. In our paper, a L-shaped six-array antenna is specially designed for spoofing interference. In the real environment, the method we designed can realize the simultaneous capture and tracking of the normal and interference signals of the same satellite through multimodal monitoring and effectively separate the real satellite and spoofing interference signals by using the airspace identification method, followed by the long- and short-baseline interferometric-direction-finding method of the spoofing interference source to locate the direction of the interference source through the designed antenna array" are added in section 1.

Reviewer 3 Report

Overall, the topic of this paper is interesting where the six-array spoofing interference monitoring array antenna is studies. There are three improvements need to be done before publishing:

1.      The motivation needs to be clarified more clearly. The essence of the proposed method in the paper is to deal with the unmodeled errors or abnormal signals.

2.      The improvements of the proposed method need to be discussed more comprehensively.

3.      The language needs to be modified carefully.

Author Response

Response to Reviewer 3 Comments

Point 1:

The motivation needs to be clarified more clearly. The essence of the proposed method in the paper is to deal with the unmodeled errors or abnormal signals.

Response 1:

Thank you for your considerate comments and suggestions. Our previous description was not clear. Currently GNSS civil navigation signals are more susceptible to interference by GNSS navigation spoofing signal generators, because they are not en-crypted like military signals. The current research results are carried out in the simulation environment or ideal environment, and it is not persuasive. We have completed the modification in section 1.

Point 2:

The improvements of the proposed method need to be discussed more comprehensively.

Response 2:

Our previous description was not clear. The description "Aiming at the problem of the traditional spoofing interference detection method can not locate the direction of the spoofing interference source, many scholars proposed the use of array antenna to locate the direction of the spoofing interference source, drawing on the relevant interferometer direction measurement principle, using the same signal received by different array elements of the carrier phase difference to build the observation, it can realize the direction of signal captured by the receiver. Spoof interference detection and identification can be achieved by comparing the satellite direction with the ephemeris solution. However, their methods stay in the experimental simulation stage and are not verified in the real outdoor environment. In the real environment, there will be interference from the external environment such as multipath, which has some influence on the detection of spoofing interference signals. In our paper, a L-shaped six-array antenna is specially designed for spoofing interference. In the real environment, the method we designed can realize the simultaneous capture and tracking of the normal and interference signals of the same satellite, through multimodal monitoring and effectively separate the real satellite and spoofing interference signals by using the airspace identification method, followed by the long- and short-baseline interferometric-direction-finding method of the spoofing interference source to locate the direction of the interference source through the designed antenna array" are added in line 83 for clarify.

Point 3:

The language needs to be modified carefully.

Response 3:

Thanks again for your considerate comments. Now, this paper has undergone English language editing by MDPI. The text has been checked for correct use of grammar and common technical terms, and edited to a level suitable for reporting research in a scholarly journal.

Reviewer 4 Report

This manuscript discusses using improved array antennas to detect spoofing interferences from the received GNSS signals. This work is significant in the GNSS community.

However, the contributions are not adequately clarified, and one can hardly see the novelty. For example, the antenna array has been commonly used in the GNSS signal processing for interference mitigation, making this manuscript's topic less convincing. Meanwhile, there is an evident disconnection between the introduction and the methodology. More importantly, incorrect definitions and ambiguity often occur in Section 2 (2. Materials and Methods), and the experiments and the discussions have much space to improve. In summary, this paper would not be able to be published in this journal in its current form, and I will reject it.

Some comments are provided as follows and hopefully would help improve the paper:

1.      The language needs to improve as the readers are hard to follow some points of view. For example,

    from lines 95 to 100, the authors propose an algorithm separating the spoofing and the actual signals, but the novelty is not evident, or the words are not entirely understandable.

    In line 111: “and the spoofing interference signal is only completed transponder,”?? One cannot really read it.

2.      The contributions of this research are ambiguous, so this research is uneasy about attracting readers and cannot be followed. The authors are encouraged to clarify their novel ideas in priority.

3.      In line 43, primary unintentional interferences for the ground GNSS users include multipath/NLOS effects which the authors fail to mention.

4.      From Lines 66 to 76: it is too long to read a story about spoofing in military action, which is unnecessary for an academic article.

5.    From Lines 174 to 175, the denotations are inconsistent with equation (4), e.g., \Delta_f and f.

6. More descriptions should be given to equation (10), and a reference is encouraged to add. The case is the same as equations (12) and (13). 

7.      In general, the spoofing signal amplitude can be much higher than the real one, which will cause a near-far effect. Will the proposed algorithm be affected by this near-far problem? If so, how to cope with it?

Round 2

Reviewer 1 Report

English has been deeply improved. L70. This reviewer would appreciate one or more little examples of the Margarita et al study "only applicable to some civil systems". Fig1. From the explanation and the figure 1, R is the Real position of the target and F is the Fake position "sent" to the target from J that is the "spoofing source". If I'm right, the S1-S4 position arrow signals should be directed to R instead of F, because F is a Fake point .. There is no target there. Is it correct or I'm wrong?. In my opinion, the "spoofing point", the "spoofing destination" are not clear. I would suggest to use always the same term, for example, "spoofing source" for J, "spoofing receiver" for R and "spoofing point" for F, or similar. This would help enormously to understand the rationale. Nevertheless, the actual version is much clearer than the first one. L131. Please replace GPS by GNSS. L132. Do you mean that \rho_i is the same in eq.1 and eq.2?. L133. Does it mean that the distance between F and J should be greater than the distance between J and R in order to do spoofing at R to F? Fig2. Please replace GPS by GNSS. I would review the whole manuscript to replace completely both terms except when GPS is required. Fig2. The figure is for the element J? ... please clarify. Fig4. Again, GNSS in place of GPS.

Reviewer 4 Report

Although the response to the 7th point does not entirely convince me, I guess the authors have made great efforts to improve this manuscript. I appreciate it a lot. I think it can be accepted as it is.

Congratulations on your great work! 

Author Response

Thank you for your affirmation of our work, and we will continue to investigate spoofing interference in the future.